# The NANOG Transcription Factor Induces Type 2 Deiodinase Expression and Regulates the Intracellular Activation of Thyroid Hormone in Keratinocyte Carcinomas

**DOI:** 10.3390/cancers12030715

**Published:** 2020-03-18

**Authors:** Annarita Nappi, Emery Di Cicco, Caterina Miro, Annunziata Gaetana Cicatiello, Serena Sagliocchi, Giuseppina Mancino, Raffaele Ambrosio, Cristina Luongo, Daniela Di Girolamo, Maria Angela De Stefano, Tommaso Porcelli, Mariano Stornaiuolo, Monica Dentice

**Affiliations:** 1Department of Clinical Medicine and Surgery, University of Naples “Federico II”, 80131 Naples, Italy; annarita.nappi@unina.it (A.N.); emery2304@gmail.com (E.D.C.); caterina.miro@unina.it (C.M.); nunziacicatiello3011@gmail.com (A.G.C.); ssagliocchi@gmail.com (S.S.); man.giusy@gmail.com (G.M.); m.angeladestefano@gmail.com (M.A.D.S.); 2IRCCS SDN, 80143 Naples, Italy; ambrosioraf.ra@gmail.com; 3Department of Public Health, University of Naples “Federico II”, 80131 Naples, Italy; cristinaluongo@gmail.com (C.L.); danieladigirolamo1988@gmail.com (D.D.G.); tommasoporcelli@gmail.com (T.P.); 4Department of Pharmacy, University of Naples Federico II. Via Montesano 49, 80149 Naples, Italy; mariano.stornaiuolo@gmail.com; 5CEINGE–Biotecnologie Avanzate Scarl, 80131 Naples, Italy

**Keywords:** thyroid hormone, deiodinases, skin cancer

## Abstract

Type 2 deiodinase (D2), the principal activator of thyroid hormone (TH) signaling in target tissues, is expressed in cutaneous squamous cell carcinomas (SCCs) during late tumorigenesis, and its repression attenuates the invasiveness and metastatic spread of SCC. Although D2 plays multiple roles in cancer progression, nothing is known about the mechanisms regulating D2 in cancer. To address this issue, we investigated putative upstream regulators of D2 in keratinocyte carcinomas. We found that the expression of D2 in SCC cells is positively regulated by the NANOG transcription factor, whose expression, besides being causally linked to embryonic stemness, is associated with many human cancers. We also found that NANOG binds to the D2 promoter and enhances D2 transcription. Notably, blockage of D2 activity reduced NANOG-induced cell migration as well as the expression of key genes involved in epithelial–mesenchymal transition in SCC cells. In conclusion, our study reveals a link among endogenous endocrine regulators of cancer, thyroid hormone and its activating enzyme, and the NANOG regulator of cancer biology. These findings could provide the basis for the development of TH inhibitors as context-dependent anti-tumor agents.

## 1. Introduction

Keratinocyte carcinoma (KC), also known as non-melanoma skin cancer (NMSC), is the most common malignancy worldwide [1]. It comprises basal cell carcinoma (BCC), squamous cell carcinoma (SCC), and actinic keratosis, all of which derive from skin keratinocytes [1]. BCCs rarely metastasize. However, as they grow, BCCs often destroy underlying tissues and should therefore be promptly treated or removed [2]. Cutaneous squamous cell carcinoma (cSCC) is moderately invasive and associated with a substantial risk of local recurrence, metastasis, and death [3].

Aberrant activation of the Hedgehog pathway is a pivotal defect implicated in BCC, and mutations in Ptch-1 (the Shh receptor) are the leading cause of the inherited form of basal cell nevus syndrome also known as Gorlin syndrome, in which patients may develop hundreds of BCCs [4,5]. Human SCCs are primarily induced by sunlight exposure; however, multistage carcinogenesis can be induced in mice using either UV irradiation or chemical carcinogens in combination with tumor promoters [6]. Mutations in Ha-RAS are a molecular fingerprint of SCC, and RAS overexpression, which phenocopies oncogenic RAS, is observed in most human cutaneous SCCs and SCC precursors [7].

Thyroid hormones (THs, T4 and T3) are key endocrine regulators of several biologic functions [8]. Their primary mode of action involves binding to the TH receptors in the nucleus and imposing a signature of gene expression that activates or inhibits the transcription of target genes [9]. Intracellular TH signaling does not faithfully reflect TH levels in plasma, rather, it is adapted inside the cell where TH signaling is modified via the action of deiodinases. Type 2 deiodinase (D2) confers on cells the capacity to produce extra amounts of T3 and thus enhance TH signaling [8]. In contrast, D3 expression results in the opposite action. D3 is a key component of epidermal homeostasis and development [10] and is overexpressed in BCCs under the control of Shh [11,12,13,14] and of the cancer-associated microRNA-21 [15,16]. We recently demonstrated that D2 is expressed also in skin cancer, in both mouse and human BCC and SCC [17,18]. By increasing the intracellular activation of T3, D2 enhances SCC cell invasiveness, epithelial–mesenchymal transition (EMT), and malignant transformation [18]. However, the mechanisms regulating the expression of D2 in skin cancer have yet to be fully explored.

NANOG is a homeodomain protein involved in the maintenance and self-renewal of embryonic stem cells [19,20]. It is highly expressed in the inner cell mass of mouse and human blastocysts and in embryonic stem cells [19] and rarely expressed in adult human tissues [21]. In the skin, NANOG has been detected in the basal cells of the mouse stratified epithelia [22]. Notably, besides its role in stemness maintenance, NANOG is expressed in multiple human cancers, namely, brain, colon, prostate, and stomach cancers [23,24,25].

Here, we report that in KC, NANOG promotes the transcription of the *Dio2* gene coding for the D2 protein. By binding the *Dio2* promoter, NANOG induces D2 transcription and enhances the intracellular activation of TH. In BCC, D2 and NANOG co-localize at the very early stages of tumorigenesis, whereas in SCC they co-localize in the late stages of tumor progression. Notably, D2 and NANOG are specifically expressed in CD34-negative cells (in the bulk of the tumor) and not in the cancer stem cell population. As already observed for D2, NANOG promotes SCC cells invasion and exacerbates tumor invasiveness [26]. Taken together, our data reveal that NANOG regulates TH metabolism and that the NANOG-dependent D2 expression is a pivotal transcriptional network exacerbating the expression of invasiveness genes in skin cancer.

## 2. Materials and Methods

### 2.1. Analysis of Transition Factor Binding Sites

We searched the murine *Dio2* promoter for putative transcription factor binding sites (TFBSs) using MatInspector (ver. 9.1, Precigen Bioinformatics, München, Germany). Binding sites were compared with those in the database, in terms of core and matrix similarity scores 0.90 (maximum 1.00). Position analyses of common TFBSs were performed in Excel and reported in Appendix A.

### 2.2. Cell Cultures and Transfections

G2N2C cells were derived from transgenic mice expressing a constitutively active form of Gli2 under the control of the keratin 5 promoter [27] and isolated from trichoblastomas. The latter are BCC-like tumors, and their corresponding cells are referred to as ‘‘BCC cells’’ throughout this article. G2N2C cells were cultured in low-calcium medium, with 8% Ca^2+^-chelated fetal bovine serum (FBS) and keratinocyte growth factor (KGF, 1.0 ng/mL) (Sigma-Aldrich, St. Louis, MO, USA). SCC-13 cells, derived from a skin SCC [28], were cultured in Keratinocyte-SFM (KSFM 1X) serum-free medium [+] L-Glu (Gibco, Thermo Fisher Scientific, Waltham, MA, USA) with bovine pituitary extract (30.0 μg/mL) and human recombinant epidermal growth factor (0.24 ng/mL). Transient transfections were performed using Lipofectamine-2000 (Invitrogen™, Carlsbad, CA, USA) for BCC cells and Lipofectamine-3000 (Invitrogen™) for SCC cells, according to the manufacturer’s instructions.

### 2.3. Short Hairpin RNA-Mediated Knock-Down of NANOG

BCC cells were grown in p60 plates until they reached 60% confluence and then transfected with shRNA targeting endogenous NANOG (shNANOG-1 and 2) or a scramble shRNA (shCTR) as a negative control using Lipofectamine-2000 (Invitrogen™). shRNA-targeted NANOG sequences were selected from the BLOCK-iT™ RNAi Designer (Invitrogen™, Thermo Fisher Scientific). Oligos were designed and ordered from Eurofins Genomics. shRNA oligos were cloned into the *EcoRI* and *XhoI* sites of a pcRNAi plasmid. The cloning steps are schematically shown in Appendix A. Forty-eight hours after transfection, total protein lysate was collected and analyzed by Western Blot, and total RNA was extracted and analyzed by real-time PCR. The sequences of oligonucleotides used for real-time PCR are shown in Appendix A. The list of the antibodies used is shown in Table 1.

### 2.4. Wound Scratch Assay

BCC and SCC cells were seeded in p60 plates until they reached 60% confluence and then transfected with a NANOG plasmid or a negative control plasmid, i.e., CMV-FLAG. Twenty-four hours after transfection, cells were treated with Mitomycin C from *Streptomyces caespitotus* (0.5 mg/mL). At T0, a cross-shaped scratch was made on the cell monolayer with the tip of a sterile 2.0 μL micropipette. The FBS-free culture medium was then replaced with fresh medium to wash out the released cells. Cell migration was measured by comparing images taken at the beginning and end of the experiment at the times indicated in each experiment, using an IX51 Olympus microscope at 10× magnification and the Cell*F Olympus Imaging Software (Olympus Corporation, Center Valley, PA, USA). ImageJ software (National Institutes of Health (NIH) Image, 9000 Rockville Pike, Bethesda, Maryland 20892) was used to draw the cell-free region limits in each case. The initial cell-free surface was subtracted from the endpoint cell-free surface and plotted in a graph, as shown in Figure 5B and Appendix A.

### 2.5. Animals, Histology, and Immunostaining

K14Cre^+/−-ERT^/Rosa-SmoM2-YFP/D2-3X-Flag (K14Cre-SMO) mice were obtained by crossing the keratinocyte-specific conditional K14Cre^+/−-ERT^/Rosa-SmoM2-YFP mice with the D2-3X-Flag mice. The expression of a constitutively active Smoothened mutant (SmoM2) in the adult epidermal-specific compartment was induced by treatment with 10.0 mg tamoxifen. All animal experiments and mouse husbandry were carried out in the animal facility of CEINGE − Biotecnologie Avanzate, Naples, Italy, in accordance with institutional guidelines (Authorization n. 354/2019-PR by the Ministero della Salute).

For immunofluorescence and histology, ears from a K14Cre-SMO mouse were embedded in paraffin, cut into 7.0 μm sections, and hematoxylin-and-eosin (H & E)-stained. Slides were baked at 37 °C, deparaffinized by xylene, dehydrated with ethanol, rehydrated in phosphate-buffered solution (PBS), and permeabilized by placing them in 0.2% Triton X-100 in PBS. Antigens were retrieved by incubation in 0.1 M citrate buffer (pH 6.0) or 0.5 M Tris buffer (pH 8.0) at 95 °C for 5 min. Sections were blocked in 1% BSA/0.02% Tween/PBS for 1 hour at room temperature, then incubated with primary antibodies overnight at 4 °C in blocking buffer, washed with 0.2% Tween/PBS, incubated with secondary antibodies at room temperature for 1 hour, and finally washed with 0.2% Tween/PBS. Images were acquired with a Leica DMi8 microscope and the Leica Application Suite LAS X Imaging Software (Leica Microsystems, Wetzlar, Germany).

### 2.6. Isolation of CSCs

Dorsal and tail skins were separated from the dorsal and tail bone and incubated overnight in 0.25× trypsin at 4 °C. The next day, the epidermis was separated from the dermis and incubated for 15 minutes in 0.25× trypsin. Trypsin was neutralized by adding DMEM containing 10% FBS. The cell suspension was then passed twice through a 70.0 µm cell strainer. Isolated keratinocytes were immunostained with APC–anti-mouse CD34 antibody (code 119310, BioLegend, San Diego, CA, USA) and rat PE–anti-human α6-integrin (CD49f; code 555736, BD Pharmingen™) by incubation for 45 minutes at room temperature. Fluorescence-activated cell sorting analysis was performed using FACS Canto2 software (FACS Canto2, Becton Dickinson, BD Biosciences-US San Jose - CA, USA). Live SmoM2-expressing epidermal cells were gated by forward scatter, by side scatter by the expression of SmoM2-YFP. For RNA analysis, sorted cells were harvested directly into TRIzol reagent (Invitrogen™).

### 2.7. HPLC–MS Measurement of T3, T4, and rT3

Standard stock solutions of all target analytes (3,3′,5,5′-tetraiodo-L-thyronine (L-thyroxine (T4), 3,3′,5-triiodothyroxine (T3), and 3,3′,5′-triiodothyronine (reverse T3, rT3) were prepared in methanol. Dilutions of each standard were prepared in methanol/water (*v*/*v*, 50/50). Ten milliliters of cell medium was deproteinated using 9 volumes of cold acetone and then centrifuged at 14,000 r.p.m. The supernatants were reduced to 200 μL under N_2_ for instrumental analysis. The HPLC system Jasco Extrema LC-4000 system (Jasco Inc., Ithaca, NY, USA) was coupled to an Advion Expression mass spectrometer (Advion Inc., Ithaca, NY, USA) equipped with an ESI source. We used 10 mM ammonia acetate in deionized water as the aqueous mobile phase and 0.1% acetic acid in methanol as the organic mobile phase. The analyses were performed in the positive Electrospray ionization (ESI) mode. Six replicates were run for each sample.

### 2.8. Statistical Analyses

All results are reported as means ± SD. Differences between samples were assessed with the Student’s two-tailed *t* test for independent samples. Relative mRNA levels (in which the control sample was arbitrarily set as 1) are reported as results of real-time PCR in which the expression of cyclophilin A served as a housekeeping gene. In all experiments, differences were considered significant when *p* was less than 0.05. Asterisks indicate significance at * *p* < 0.05, ** *p* < 0.01, and *** *p* < 0.001 throughout.

## 3. Results

### 3.1. NANOG Induces the Transcription of Type 2 Deiodinase

To investigate the molecular mechanisms that regulate D2 expression in skin tumorigenesis, we analyzed a region of 1.3 kb upstream of the annotated transcription start site of the *Dio2* gene using the MatInspector software (Precigen Bioinformatics, München, Germany). The matrix data of each sequence are reported in Appendix A. The in silico analysis of the KEGG pathway revealed that 56 genes were putative transcription factors that regulate D2 in cancer (Figure 1A). Among them, a NANOG binding site was present at position −249 to −231 bp, with a matrix similarity of 0.94 (Figure 1B). The sequence of the NANOG binding site is the highly conserved catcaattCATTcaattcc motif (uppercase letters refer to the core sequence).

To investigate the mechanisms whereby NANOG regulates D2 expression, we ectopically expressed NANOG in a murine BCC cell line, i.e., the G2N2C cell line [27]. NANOG induced the expression of D2 mRNA (Figure 1C) and the activity of the murine Dio2-LUC promoter (Figure 1D). Importantly, a chromatin immunoprecipitation assay in BCC cells confirmed that NANOG physically binds the *Dio2* promoter (Figure 1E). Moreover, overexpression of NANOG increased the expression of D2 protein measured by immunofluorescence and Western Blot analysis in primary cultures of mouse keratinocytes from a D2-Flag mouse (used to overcome the lack of commercially available functional D2 antibodies, Figure 1F,G and Appendix A). Overall, these results indicate that NANOG positively regulates *Dio2* gene expression.

To assess whether the NANOG-mediated D2 induction results in enhanced intracellular TH activation, we measured the amount of TH in the BCC culture medium after transfection with NANOG. HPLC–MS analysis demonstrated that T3 secreted in the culture medium was increased by NANOG overexpression (Figure 2A). Accordingly, NANOG overexpression also increased the intracellular TH action as observed by the co-transfection of increasing amounts of NANOG together with an artificial T3-responsive promoter, namely, TRE3-TK-LUC (Figure 2B).

Next, the core motif of NANOG was mutagenized within the *Dio2* promoter region, generating a *Dio2*-responsive promoter mutant for the NANOG binding site, namely, the Dio2mut-LUC promoter (Figure 2C and Appendix A). When we transfected the wild-type and the mutant D2 promoters, we found that mutagenesis of the NANOG binding site strongly reduced the basal *Dio2* promoter activity (Figure 2D, left). Moreover, mutation of the NANOG binding site on *Dio2* promoter potently reduced the upregulation of D2 induced by ectopic transfection of NANOG (Figure 2D, right). Finally, to further test the role of NANOG in D2 upregulation, we inhibited NANOG in BCC cells by shRNA-mediated NANOG silencing (Appendix A). In accordance with its role as an inducer of D2 expression, NANOG downregulation resulted in a decrease of D2 expression at mRNA and promoter activity levels (Figure 2E,F).

### 3.2. NANOG and Type 2 Deiodinase Colocalize in Mouse BCC and Are Expressed in the Early Phases of Tumorigenesis

Having established that NANOG regulates D2 expression in BCC cells in vitro, we asked whether their expression correlated in genetically induced BCC tumors in vivo. To this aim, we used an inducible mouse model that enables the expression of a constitutively active Smoothened (SmoM2) in the adult epidermal compartment [29] and presents BCC-like tumors, i.e., the K14Cre^ERT^/Rosa-SmoM2-YFP/D2-3xFlag mouse model [30,31,32] (herein referred to as K14Cre-SMO). In this mouse model, in which the endogenous gene coding for D2 is fused to a Flag tag, BCCs are induced by the activation of the Smo oncogene in the basal cells of the epidermis by tamoxifen administration. We observed that while D2 was expressed at very low levels in normal skin, its expression rapidly increased in the early stages of tumorigenesis (1–2 weeks after tumor induction) and declined thereafter (Figure 3A). Staining with an anti-Flag antibody confirmed that D2 was specifically expressed in the early stages of BCC formation, i.e., two weeks after tamoxifen-induced Smo expression (Figure 3B). Notably, NANOG expression paralleled D2 expression. Indeed, NANOG was rapidly upregulated during the early steps of BCC initiation and dramatically decreased some weeks later (Figure 3C). Immunofluorescence analysis confirmed the transcriptional profile of D2 and NANOG during BCC tumorigenesis and revealed co-expression of D2 and NANOG very early in tumorigenesis (Figure 3D). The latter results indicate that the NANOG–D2 axis plays a stage-specific role in BCC development, i.e., NANOG and D2 are required for BCC initiation but not for BCC progression.

Since NANOG is a crucial regulator of stemness, we asked if D2/NANOG co-expression targets both the bulk of tumor cells and the cancer stem cell (CSCs) population. To this aim, we isolated the CSCs by fluorescence-activated cell sorting from K14Cre-SMO mice (by isolating the triple-positive, SmoM2-YFP^+^/α_6_-integrin^+^/CD34^+^ cells) at different time points after tamoxifen administration (Figure 3E). This strategy discriminates among the total population of cancer cells (SmoM2-YFP^+^/α_6_-Integrin^+^ cells), the CSCs (CD34^+^ cells), and the somatic cells of the bulk of the tumor (CD34^−^ cells). The CD34^+^ cells have the features of adult epidermal cancer stem cells, as reported by Youssef and colleagues [32] and as confirmed by us by measuring a panel of markers of epidermal CSCs (Sox2, Sox9, LGR5 and NRG1, Appendix A). Intriguingly, in the CSCs (CD34^+^ cells), D2 and NANOG were expressed at time 0 (normal skin), and their expression drastically decreased upon tumorigenic induction (Figure 3F,H).

On the contrary, the non-CSC population (i.e., CD34^−^ cells), which constitutes the bulk of the tumor, confirmed a high D2 and NANOG expression level at the early stages of tumorigenesis, one and two weeks following tumor induction (Figure 3G). Therefore, we concluded that D2 and NANOG are specifically expressed in the tumor epithelium of BCC and not in the CD34^+^ population of CSCs and that the expression of D2 and NANOG in CSCs drastically differs from that in the bulk of tumors.

### 3.3. NANOG and Type 2 Deiodinase Expression Is Associated with Advanced SCC Tumors in Mouse and Humans

To determine whether D2 and NANOG co-expression is BCC-specific or if it is present in other KCs, we used an alternative model of KC, namely, SCCs (Figure 4A). In agreement with our previous finding [18], we observed that D2 expression in SCCs is activated during the late phase of tumorigenesis and peaks during the final phases of SCC tumorigenesis (Figure 4B). Importantly, NANOG expression was also up-regulated during the late stages of tumor progression, when papillomas become more invasive and turn into SCCs (Figure 4B). We also evaluated the expression of D2 and NANOG in human SCC tumors at different pathologic stages. Importantly, human D2, NANOG (encoded by the *NANOG1* gene), and NANOG-P8 (a transcribed retrogene of the human *NANOG* homeobox gene [22,33]) were all highly expressed in the late stages of human tumorigenesis, which is consistent with their expression during mouse tumor progression (Figure 4C–E). Pearson’s correlation analysis revealed a direct significant correlation between D2 and NANOG and between D2 and NANOG-P8 in each human SCC sample (Figure 4F: D2 versus NANOG R = 0.868, p = 0.000011; Figure 4G: D2 versus NANOG-P8: R = 0.332, *p* = 0.017). These results confirm the close link between D2 and NANOG in human SCC tumors.

### 3.4. The Intracellular Activation of TH Is Critical for the Oncogenic Effects of NANOG

To assess the functional relevance of the newly identified NANOG–D2–TH axis, we assessed the consequences of TH attenuation in the oncogenic action of NANOG. To this aim, we overexpressed NANOG and inhibited D2 in BCC cells, thereby measuring the proliferation, migration, and invasiveness of BCC cells. D2 was inhibited by treating cells with rT3, which is a competitive and specific inhibitor of D2 activity [34,35]. As an additional control of TH signal inhibition, we cultured cells in a TH-stripped medium (charcoal-stripped, CH) and transiently transfected them with NANOG. While proliferation was not affected by NANOG overexpression (Appendix A), cell migration was potently upregulated by NANOG, but not in the presence of rT3 or in charcoal-stripped medium (Figure 5A,B). The expression of the mesenchymal markers N-cadherin, vimentin, and ZEB-1 was increased, while the E-cadherin/N-cadherin ratio was reduced at both mRNA and protein levels by NANOG, thus confirming that NANOG induces the EMT program (Figure 5C,D, Appendix A). Importantly, the inhibition of D2 by rT3 drastically reduced the NANOG-dependent EMT (Figure 5C,D, Appendix A). Similarly, in SCC cells, D2 inhibition and CH medium attenuated the NANOG-dependent acceleration of cell migration (Appendix A) and the expression of EMT markers (Appendix A). Altogether, these data set the stage for the use of hormonal regulation as a tool with which to manipulate the homeostasis of cancer cells and thus to modulate their expansion and differentiation in a therapeutic context.

## 4. Discussion

The homeobox domain transcription factor NANOG, which is a key regulator of embryonic development and cellular reprogramming, has been reported to be broadly expressed in human cancers and exert a pro-tumorigenic effect [36]. We recently demonstrated that the thyroid hormone-producing enzyme D2 is highly expressed in the advanced stages of human and mouse cutaneous SCC [18]. Our previous in vitro and in vivo data showed that in SCC, D2-produced T3 enhances tumor progression, while D2 inhibition reduces the metastatic potential of SCCs. The molecular mode of T3-mediated invasiveness primarily involves the transcription of the master regulator of EMT, ZEB-1, and the induction of the downstream genes for vimentin, N-cadherin, and several metalloproteases [18]. However, the regulation of D2 in cancer has not yet been addressed.

In the present study, we describe a novel regulation loop in which NANOG positively regulates the D2 enzyme and induces the upregulation of thyroid hormone signaling in BCC and SCC cells. In this model, NANOG and D2 play stage-specific roles in KCs. In the relatively benign BCC tumors, the two proteins are expressed only in the early phases of tumorigenesis. In BCC cells, D2 and NANOG increase cell migration ability, without affecting cell proliferation, thereby supporting the hypothesis that the NANOG–D2–TH axis increases the migration of tumoral cells toward the derma. In the more aggressive SCC tumors, NANOG and D2 are expressed at the late stages of tumorigenesis, during which they support invasiveness and EMT. Common downstream target genes code for ZEB-1, vimentin, E-cadherin, and metalloproteases, thereby resulting in enhanced cell migration and mesenchymal phenotype.

NANOG (*NANOG1*) has 11 pseudogenes, among which *NANOGP4*, *P5*, and *P8* are expressed in cancer cells [37]. Notably, only *NANOGP8*, which is expressed in humans and not in mice, encodes a NANOG protein that differs from NANOG1 only in a single, conserved amino acid [24]. Therefore, the NANOG protein can be coded by two different genes in human cancer cells [24], while only one NANOG protein is expressed in mouse cancer cells.

How NANOG functions in cancer is still a matter of debate, and its function in cancer cells seems to be highly context-dependent [38]. In various cancer types, NANOG exerts a pro-tumorigenic action associated with either cancer growth promotion, as observed in human gastric cancer cells [39], glioblastomas, and breast cancer cells [25], or enhanced invasiveness and malignancy, as observed in breast cancer and hepatic tumors [40,41]. In cutaneous SCC cells, NANOG induces EMT and malignant transformation, thereby worsening the malignant phenotype of SCC tumors [26].

It is not known whether NANOG expression is restricted to CSCs or whether it is also expressed in the tumoral mass of somatic cells [36]. Unexpectedly, our data show that D2 and NANOG are highly expressed in the cells of the tumoral bulk (CD34^−^ cells), while their expression is extremely low in the stem cell population (CD34^+^ cells). This finding suggests that NANOG plays a specific cancer-related role in tumoral cells that is not associated with its ability to induce stemness. Although this finding seems to be in contrast with NANOG as a stem gene, it is in line with the finding that NANOG is expressed in adult stratified epithelia, particularly, in the esophagus, forestomach, skin, urothelium, and mucosal tissue [22], which indicates that this transcriptional factor plays a more complex role in somatic cells than hitherto believed.

A novel finding of our study is that NANOG is a positive regulator of TH signaling. Indeed, by inducing D2 expression, NANOG upregulates the production of the active TH (T3) in the target cells (Figure 2A,B). Our data indicate that in SCC cells, this represents a forward loop that drives the cancer cellular phenotype toward the mesenchymal state. Thus, one effect of inducing NANOG expression is to activate the T4-to-T3 conversion by D2, thereby activating a downstream EMT axis. In this context, we found that deprivation of TH from the culture medium or block of D2 enzymatic activity inhibited EMT induced by NANOG. This finding is consistent with the finding that D2 and NANOG expression is associated with a worse SCC prognosis [18,26] and may belong to the molecular machinery of tumor progression.

## 5. Conclusions

In summary, we described a hitherto unknown function of the NANOG protein, namely, that it induces TH signaling activation in skin tumors. Our data indicate that the NANOG–D2 axis described herein is critical in dictating the oncogenic potential of NANOG and reinforce the role of D2 as a positive regulator of tumor invasiveness and EMT in epithelial cancers. These considerations indicate that the induction of EMT in KCs is a centrally important mechanism activated by TH for the progression of carcinomas to metastatic stages. However, many steps of this mechanistic model still require experimental explanations. Moreover, it remains unclear at present whether NANOG–D2 regulation is related only to epithelial cancer cells or is effective in controlling TH signaling in many other patho-physiological contexts.

## Figures and Tables

**Figure 1 cancers-12-00715-f001:**
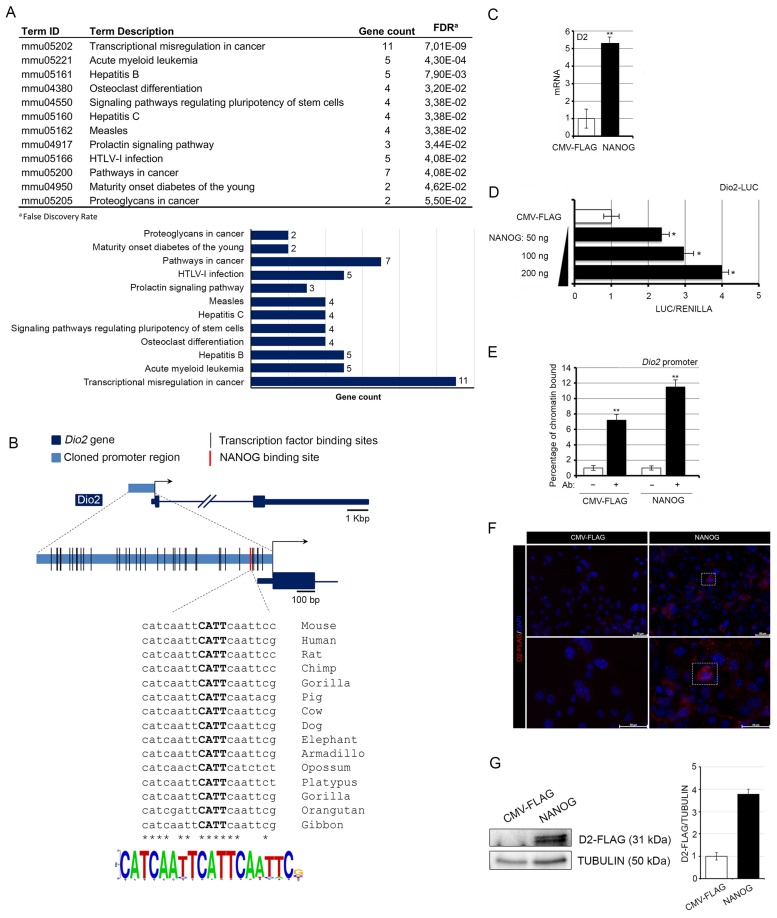
NANOG positively regulates type 2 deiodinase (D2) transcription. (**A**) KEGG pathway clusters generated by the in silico analysis of the *Dio2* promoter region. (**B**) Schematic localization of transcription factors and of the NANOG binding site within the *Dio2* promoter region; conservation and Logo representation of the NANOG binding motif. (**C**) D2 mRNA expression was measured by real-time PCR in basal cell carcinoma (BCC) cells transiently transfected with a NANOG-expressing vector or the CMV-FLAG plasmid (control). (**D**) BCC cells were transiently transfected with the Dio2–LUC promoter and with increasing amounts of the NANOG plasmid. Cells were harvested 48 h after transfection and analyzed for luciferase activity. CMV-Renilla was co-transfected as an internal control. The results are shown as means ± SD of the LUC/Renilla ratio from at least three separate experiments, performed in triplicate; * *p* < 0.05, ** *p* < 0.01. (**E**) Chromatin immunoprecipitation assay was performed in BCC cells. Immunoprecipitation of chromatin using the anti-NANOG antibody revealed that the *Dio2* gene is a direct target of NANOG. (**F**) D2-FLAG expression levels were measured by immunofluorescence analysis of mouse primary keratinocytes from D2-Flag mice transfected with the NANOG plasmid or the CMV-FLAG plasmid. Magnification 10× and 20×; scale bars represents 50 μm. (**G**) Western Blot analysis of D2 expression in the same cells as in F.

**Figure 2 cancers-12-00715-f002:**
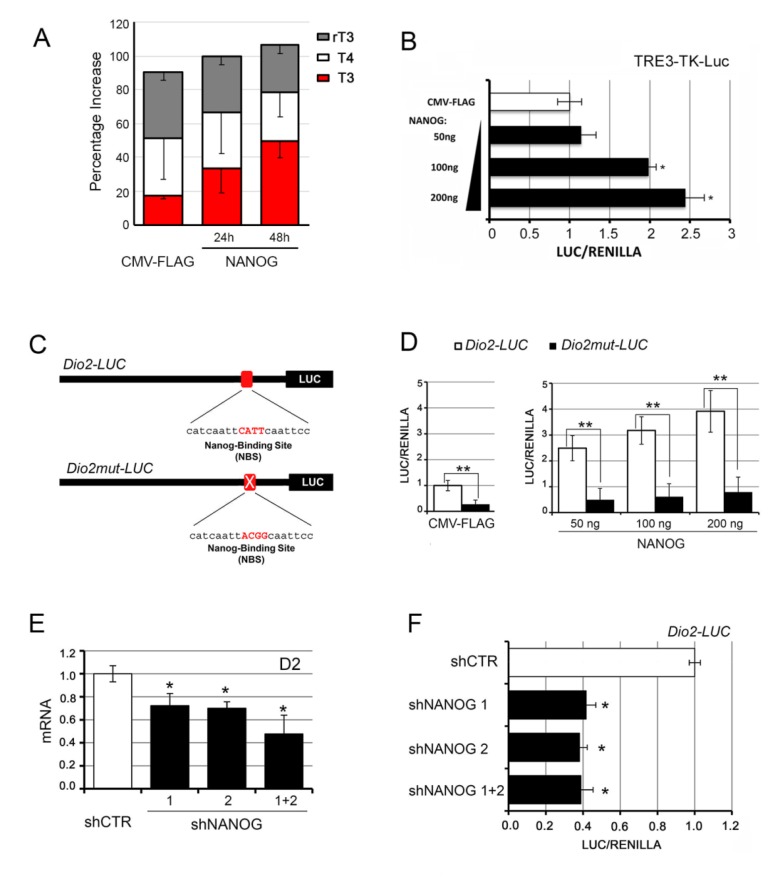
NANOG downregulation reduces the expression of D2 in BCC cells. (**A**) Levels of T3, T4, and rT3 measured by HPLC–MS in the culture medium of BCC cells transiently transfected with NANOG. Values are expressed as percentage of total thyroid hormones (THs). (**B**) BCC cells were transiently transfected with the TRE3-TK-LUC promoter and with increasing amounts of the NANOG plasmid. Cells were harvested 48 h after transfection and analyzed for luciferase activity. CMV-Renilla was co-transfected as an internal control. The results are shown as means ± SD of the LUC/Renilla ratio from at least three separate experiments, performed in triplicate; * *p* < 0.05, ** *p* < 0.01. (**C**) Schematic representation of the NANOG binding site mutation within the *Dio2* promoter region. (**D**) BCC cells were transiently transfected with Dio2-LUC promoter or Dio2mut-LUC promoter and CMV-FLAG (left) or increasing amounts of NANOG plasmid (right). Cells were harvested 48 h after transfection and analyzed for luciferase activity. CMV-Renilla was co-transfected as an internal control. The results are shown as means ± SD of the LUC/Renilla ratio from at least three separate experiments, performed in triplicate; * *p* < 0.05, ** *p* < 0.01. (**E**) D2 mRNA expression was measured by real-time PCR in BCC cells transfected with two different NANOG shRNAs, as single plasmids or in combination, or a control (CTR) shRNA as indicated. (**F**) BCC cells were transiently transfected with Dio2-LUC promoter and with NANOG shRNA as in E. Cells were harvested 48 h after transfection and analyzed for luciferase activity. CMV-Renilla was co-transfected as an internal control. The results are shown as means ± SD of the LUC/Renilla ratio from at least three separate experiments, performed in triplicate; * *p* < 0.05, ** *p* < 0.01.

**Figure 3 cancers-12-00715-f003:**
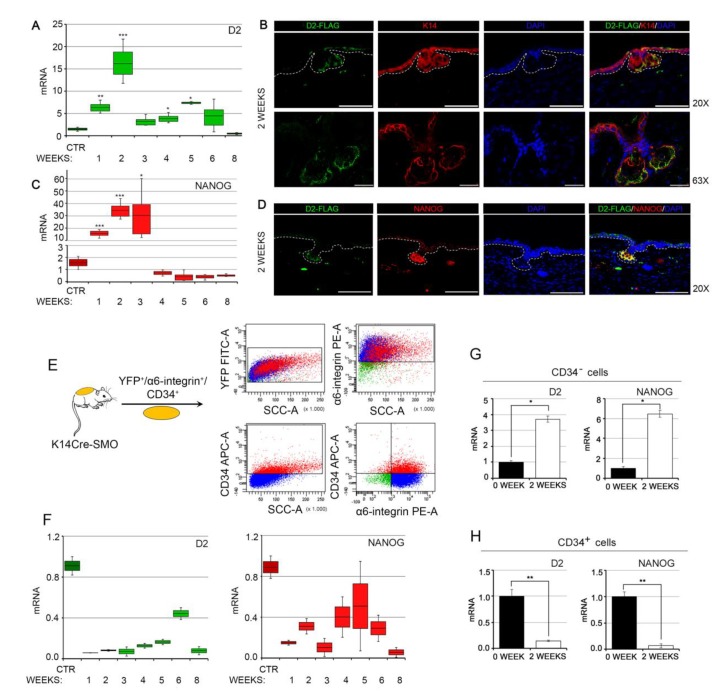
D2 expression correlates with NANOG expression during BCC tumor formation. (**A**) D2 expression was assessed by real-time PCR at different stages of BCC tumorigenesis in the adult epidermis of K14Cre-SMO mice. (**B**) D2 localization was assessed by immunofluorescence analysis two weeks after the induction of BCC tumorigenesis in the adult epidermis of K14Cre-SMO mice. Magnification 20× and 63×; scale bars represent 100 μm and 25 μm, respectively. (**C**) Relative expression of NANOG mRNA at different stages of BCC tumorigenesis, as in A. (**D**) Representative D2/NANOG co-staining was performed on paraffin-embedded skin sections at two weeks of BCC tumorigenesis from the ear epidermis of K14Cre-SMO mice. Magnification 20×; scale bars represent 100 μm. Data represent the mean of four independent experiments performed in triplicate. (**E**) Schematic representation of the strategy used to isolate cancer stem cell (CSCs) from K14Cre-SMO. (**F**) Transcriptional profile of D2 and NANOG in the CSCs population (CD34^+^ cells) during BCC tumorigenesis. (**G**) D2 and NANOG mRNA expression levels in FACS-isolated CSCs (CD34^+^ cells) were measured by real-time PCR. (**H**) D2 and NANOG mRNA expression levels in the FACS-isolated non-CSC population (CD34^−^ cells) were measured by real-time PCR. Data represent the mean of four independent experiments performed in triplicate; * *p* < 0.05, ** *p* < 0.01, *** *p* < 0.001.

**Figure 4 cancers-12-00715-f004:**
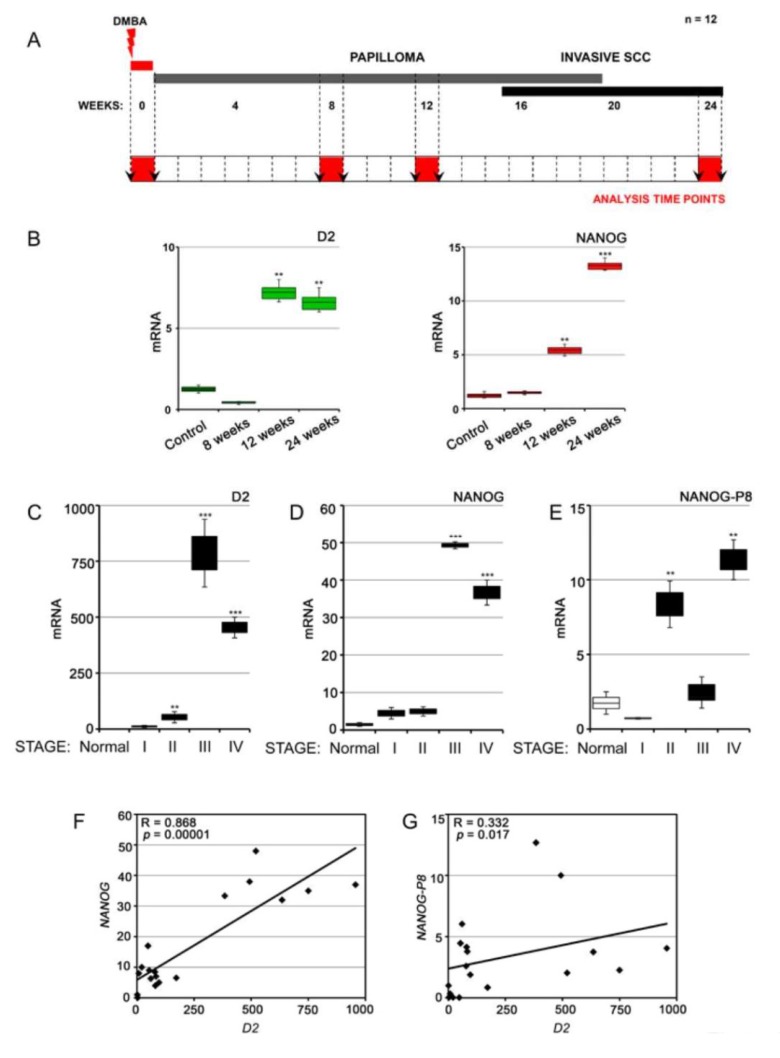
D2 and NANOG are co-expressed in the late phases of squamous cell carcinoma (SCC) tumor progression. (**A**) Schematic representation of the chemical cancerogenesis experiment for SCC tumor progression in mice. (**B**) Expression profile of D2 mRNA and NANOG mRNA during mouse SCC tumorigenesis. Data represent the mean of three independent experiments performed in triplicate. (**C**–**E**) The expression of D2, NANOG, and NANOG-P8 mRNA was measured in human SCC tumors at different pathologic stages in comparison with normal skin counterparts. All samples were run in triplicate and referred to normal skin set arbitrarily as 1. (**F**,**G**) Pearson’s correlation analysis was performed for the same data as in C–E. PCR analysis was performed in 20 different tissues for each grading. ** *p* < 0.01; *** *p* < 0.001

**Figure 5 cancers-12-00715-f005:**
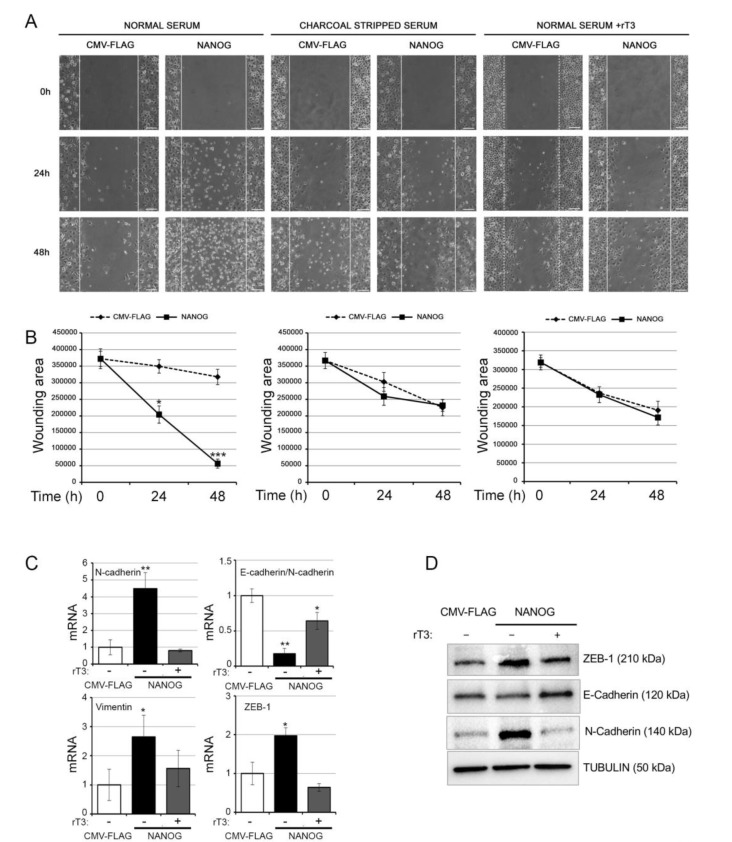
D2 inhibition attenuates the migration and mesenchymal gene expression of BCC cells induced by NANOG. (**A**) A Wound scratch assay was performed with BCC cells transfected with THE NANOG plasmid or the CMV-FLAG plasmid. The migration index was measured at 0, 24, and 48 h, under three culture conditions: (i) normal serum, (ii) charcoal-stripped serum, and (iii) normal serum + 30.0 nM rT3. Scale bars represent 100 μm. (**B**) Summary graph showing wounding area at the indicated time points during the scratch wound assay as in A. Data represent the mean of three independent experiments performed in duplicate. (**C**) N-cadherin, E-cadherin/N-cadherin ratio, vimentin, and ZEB-1 mRNA levels in BCC cells transfected with the NANOG plasmid or the CMV-FLAG plasmid and treated or not with 30.0 nM rT3. (**D**) Western Blot analysis of E-cadherin, N-cadherin, and ZEB-1 expression in the same cells as in C. Tubulin expression was measured as a loading control. Data represent the mean of three independent experiments. * *p* < 0.05; ** *p* < 0.01.

**Table 1 cancers-12-00715-t001:** List of antibodies used for molecular analyses.

Antibodies	Source	Identifier	Diluition
Anti-α-Tubulin antibody, Mouse monoclonal	Sigma-Aldrich	T8203	1:5000 WB
Anti-Cytokeratin 14, Rabbit polyclonal	Covance^®^	CLPRB-155P	1:2000 IF
Anti-Cytokeratin 17, Rabbit polyclonal	abcam	ab53707	1:2000 IF
Anti-FLAG^®^ M2 antibody, Mouse monoclonal	Sigma-Aldrich	F3165	1:1000 IF
			1:1000 WB
Anti-E-cadherin, Mouse monoclonal	BD Biosciences	610181	1:1000 WB
Anti-N-cadherin, Rabbit polyclonal	Elabscience^®^	E-AB-32170	1:500 WB
Anti-Nanog antibody, Rabbit polyclonal	abcam	ab80892	1:1000 WB
Anti-Nanog (D73G4) XP^®^, Rabbit monoclonal	Cell-Signaling	#4903	1:2000 WB
Anti-ZEB1, Rabbit polyclonal	abcam	ab155249	1:500 WB
APC ANTI-MOUSE CD34 ANTIBODY 25 UG	BIOLEGEND	119309	1:100 FACS
PE Rat Anti-Human CD49f Clone GoH3 (RUO)	BD Pharmingen™	555736	1:100 FACS

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
