# Peer review of "The NANOG Transcription Factor Induces Type 2 Deiodinase Expression and Regulates the Intracellular Activation of Thyroid Hormone in Keratinocyte Carcinomas"

_cancers, 2020, doi:10.3390/cancers12030715_

Round 1

Reviewer 1 Report

The manuscript entitled “The NANOG transcription factor induces type 2 deiodinase expression and regulates the intracellular activation of thyroid hormone in keratinocyte carcinomas”, by A. Nappi and colleagues, is a well written and well organized report showing the positive regulation of type 2 deiodinase (D2) mediated by the NANOG transcription factor. In addition, the authors analyze D2 and NANOG co-expression at different stages of tumorigenesis and indicate a role for the NANOG-D2 axis in the epithelial tumorigenesis.

Major points

Overall, this reviewer has no major concerns on this study, except for the sample size, that appears not fully described in the figure legends of the manuscript. For instance, referring to Figure 3 and Figure 4: what is the N (sample size) in the illustrated analyses?

Minor points:

Abstract

Line 23: please explain the Dio2 gene symbol

Introduction

Line 50: please introduce the use of D for deionidase(s) and explain the use of D2 for deionidase type 2.

Line 65: please explain the Dio2 gene symbol

Materials and Methods

Lines 95-96: restriction sites should be in italics

Results

Line 305: please remove the comma after “both”

References

Line 455: there is an extra line between reference 29 and 30

Supplementary material:

Figure S3: what is the sample size in the displayed experiments?

Figure S5: sample size? Duplicated or triplicated experiments?

Author Response

Major points

Overall, this reviewer has no major concerns on this study, except for the sample size, that appears not fully described in the figure legends of the manuscript. For instance, referring to Figure 3 and Figure 4: what is the N (sample size) in the illustrated analyses?

We thank the Reviewer for noting this oversight. We have added the sample size to the legends of Figures 3, 4, 5, S3 and S5.

Minor points:

Abstract

Line 23: please explain the Dio2 gene symbol.

A. The promoter is now spelled out.

Introduction

Line 50: please introduce the use of D for deionidase(s) and explain the use of D2 for deionidase type 2

A. We have corrected the text as follows: “Type 2 deiodinase (D2)”. 

Line 65: please explain the Dio2 gene symbol.

A. We now indicate the Dio2 gene symbol as the “Dio2 gene coding for the D2 protein”.

Materials and Methods

Lines 95-96: restriction sites should be in italics

A. Thank you, the restriction sites are now in italics

Results

Line 305: please remove the comma after “both”

A. We have removed the comma

References

Line 455: there is an extra line between reference 29 and 30.

A. Thank you; we have eliminated the extra line.

Supplementary material:

Figure S3: what is the sample size in the displayed experiments?

Figure S5: sample size? Duplicated or triplicated experiments?

A. We now report the sample sizes in the Figure legends of each experiment.

Reviewer 2 Report

In this study, the authors showed that NANOG directly promotes D2 transcription, so that activated TH signals could lead to skin cancer migration and EMT. This paper suggesting the importance of NANOG-D2-TH axis for skin cancer progression is considered valuable for publication. On the other hand, the following issues should be resolved before publication.

  1. Line 161: Table 1 is not found.
  2. Figure 1A and B are too small. Characters are not legible in real size.
  3. Figure 3 is small in size.
  4. It is necessary to confirm the reproducibility of the protein level in Figure 5D and test for significance.
  5. Do CSCs separated by FACS functionally have the characteristics of CSC? Experimental verification seems necessary.

Author Response

Line 161: Table 1 is not found.

A. We apologize for not uploading Table 1 in the Supplementary Data. The Table 1 has been uploaded in the revised version of the paper.

Figure 1A and B are too small. Characters are not legible in real size.

A. We have enlarged the characters on these two figures, and they are now easily legible.

Figure 3 is small in size.

A. We have enlarged this figure.

It is necessary to confirm the reproducibility of the protein level in Figure 5D and test for significance.

A. We apologize for the poor quality of the western blot in Figure 5D. To confirm the data and its significance, we repeated the experiments and ZEB1 expression is now more evident.

Do CSCs separated by FACS functionally have the characteristics of CSC? Experimental verification seems necessary.

A. To address this question, we measured the expression levels of a panel of skin stemness markers (namely, Sox2, Sox9, LGR5 and NRG1). The analysis confirmed that the CD34+ cells isolated by FACS have the classical characteristics of CSCs. This is shown in the new Figure S3 and mentioned under “Results”.